# A Learning Framework for Supporting Digital Innovation Hubs

Joao Sarraipa [1], Majid Zamiri [1,*], Elsa Marcelino-Jesus [1], Andreia Artifice [1], Ricardo Jardim-Goncalves [1] and Néjib Moalla [2]

1   NOVA School of Science and Technology, Center of Technology and Systems (UNINOVA-CTS) and Associated Lab of Intelligent Systems (LASI), NOVA University Lisbon, 2829-516 Lisbon, Portugal; jfss@uninova.pt (J.S.); ej@uninova.pt (E.M.-J.); afva@uninova.pt (A.A.); rg@uninova.pt (R.J.-G.)
2   DISP Laboratory, Université Lumière Lyon 2, 69500 Bron, France; nejib.moalla@univ-lyon2.fr
*   Correspondence: ma.zamiri@campus.fct.unl.pt

**Abstract:** With the increasing demand for digital transformation and (digital) technology transfer (TT), digital innovation hubs (DIHs) are the new piece of the puzzle of our economy and industries' landscapes. Evidence shows that DIHs can provide good opportunities to access needed innovations, technologies, and resources at a higher level than other organizations that can normally access them. However, it is critically important to note that DIHs are still evolving, under research, and under development. That is, there are many substantial aspects of DIHs that should be considered. For example, DIHs must cater to a wide spectrum of needs for TT. From this perspective, the contribution of this work is proposing a generic and flexible learning framework, aiming to assist DIHs in providing suitable education, training, and learning services that support the process of (digital) TT to companies. The proposed learning framework was designed, evaluated, and improved with the support of two EU projects, and these processes are discussed in brief. The primary and leading results gained in this way show that the learning framework has immense potential for application to similar cases, and it can facilitate and expedite the process of TT to companies. The study is concluded with some directions for future works.

**Keywords:** learning framework; digital innovation hub; technology transfer





## 1. Introduction

Digital transformation has recently attracted the attention of businesses and companies at a pace never seen before. Digital transformation—which is the integration of digital technology into all areas of a business—has become a hallmark of business empowerment [1]. In the digital sphere, digital transformation and digital technology transfer have enabled businesses and companies of all sizes to boost their competitiveness. For example, they help businesses and companies to increase their profitability, marketability, and agility; improve their operational efficiency, the quality of products and services, and collaboration within and across functions; leverage their customer satisfaction into loyalty; drive sustainability efforts; as well as reduce their costs and human error [2]. Furthermore, the processes of digital transformation and (digital) technology transfer are very important for companies as they provide access to specialized expertise, accelerate innovation, enhance competitiveness, offer cost savings, enable flexibility and adaptability, promote collaboration and partnerships, and facilitate talent development. Through embracing digital transformation and (digital) technology transfer, companies can leverage external knowledge and resources to drive growth, develop sustainability, stay ahead of the competition, and thrive in the digital economy [3–5]. On the other hand, nowadays, a large number of universities around the world focus more and more on these processes through transferring knowledge, research outcomes, and intellectual property developed within a university setting to the commercial sector or other organizations for further development, application, and utilization. The digital

transformation and transfer of technology aim to bridge the gap between academia and industry, fostering innovation, economic growth, and societal impact [6–8]. Despite offering many opportunities, the process of transferring, adapting, and integrating (digital) technologies might impose various challenges and entail different organizational and technical changes to businesses and companies [9]. To respond to this challenge, one potential solution is to take advantage of DIHs.

DIHs are supporting organizations and regional multi-partner cooperatives that assist companies—particularly traditional industry sectors, medium-sized enterprises (SMEs), start-ups, small businesses, and mid-caps—to better understand how they can improve their business, production processes, products, and services not only through incorporation of digital technologies but also by means of TT, automation, etc. In essence, DIHs attempt to provide, facilitate, and/or expedite access to the resources, innovation services, technical expertise, prototyping solutions, experimentation and testing facilities, training services, and training materials (e.g., courses, syllabuses, methods, and toolkits) required for successful digital transformation and (digital) TT. As such, DIHs can bring about commercial and financial support, connections among interested companies and investors (e.g., for financing the TT), and make a link between the suppliers and consumers of digital innovations across the value chain [10,11].

Generally, DIHs are created and built up by research and technology organizations (RTOs) and/or university labs and serve as not-for-profit organizations, service points, and one-stop shops (that provide a number of different services or might sell some products in one place). DIHs foster the mindset of developing an innovative ecosystem where all economic sectors (e.g., companies) and nonprofit organizations (e.g., public sectors, training institutes) can reap the benefits of digital investment and (digital) TT [12]. In this study, the focus of attention is on (digital) TT to economic sectors, particularly companies. DIHs encourage companies to uptake the latest advances and adopt digital technologies coming from different areas including but not limited to AI, IoT, big data, robotics, Industry 4.0, cybersecurity, photonics, and high-performance computing. Such technologies enable companies to exploit, for example, high-performance computers, e-government tools, and advanced digital skills. Given that, DIHs either independently or in collaboration and networking with other hubs, can offer several services that would not be readily accessible elsewhere [13–15]. To clarify, Figure 1 illustrates a typical DIH ecosystem. Each DIH attempts to provide a set of needed resources (e.g., human, financial) and toolkits (e.g., tools, training materials, guidance) that support the process of digital transformation and (digital) TT. In this ecosystem, the DIH might be networked and provide services collaboratively.

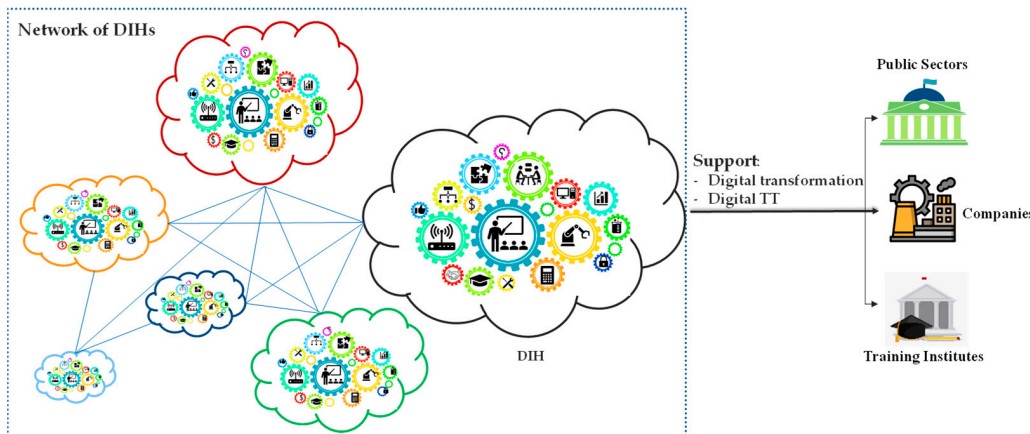

**Figure 1.** A typical DIH ecosystem.

Despite the impressive achievements made thus far in this field of study as well as the successful outcomes gained by DIHs all around the world, this body of knowledge is

still evolving. Furthermore, very few studies have already dealt with the issue of learning and training services of DIHs that support the process of (digital) TT to companies. That is, there is always a demand for more contribution, investigation, and documentation about identifying, creating, and developing the needed services and knowledge in DIHs that can properly and effectively support the TT to potential and interested companies. This work, therefore, intends to propose a generic learning framework (LF) that can be used by DIHs as a guide for creating and developing the required education, training, and learning services that facilitate the process of (digital) TT to companies. In other words, this study aims to propose some solutions that can steer the process of creating, developing, and implementing the educational, training, and learning services in DIHs which in turn lead to acquiring the knowledge needed for TT to companies.

In the following, we succinctly explain the research method used in this work and the procedures applied to gather, analyze, and interpret data in order to address our research question, hypothesis, and objective.

- Research question: How can the process of (digital) TT to companies be supported by DIHs?
- Hypothesis: The process of (digital) TT to companies can be supported by DIHs if an appropriate guiding learning framework is used.

Considering the research question and hypothesis, the main objective of this work is to propose a generic and flexible learning framework that can guide and uphold DIHs in providing suitable educational, training, and learning services that potentially could support the process of knowledge development, digital transformation, and TT to companies. Thus, in this work, we applied the hypothetical-deductive method which involves the formulation of a research question and hypothesis (addressed above) and the subsequent testing of the hypothesis through empirical observation and data analysis. In this direction, the following steps were taken:

- Deductive reasoning, as a logical approach, helped us to progress from a general idea to a specific conclusion. That is, if the proposed learning framework successfully supports a DIH to transfer digital technologies to a company, it could potentially support other DIHs and companies that follow the same objective.
- Data collection presents the methods and techniques we used for providing evidence to support the predictions derived from the hypothesis, which include published literature sources (e.g., our prior related publications) [12,15] and observations (watching and recording) of prior and existing use cases.
- Data analysis helped us to inspect, clean, transform, and interpret the data collected in the prior step. It involves applying statistical and analytical techniques (e.g., hypothesis testing, which is explained in Section 5) to understand and draw conclusions from the collected data. After conducting the analysis, the results were interpreted and translated by the authors into meaningful insights. This involves understanding the implications of the findings, drawing conclusions, and making informed decisions based on the analysis outcomes. In this step, the limitations associated with the analysis (small sample size used) were also considered.

The rest of this work is organized as follows. In Section 2, the state of the art is explained. In Section 3, we explain the methodology used for creating the LF. In Section 4, the proposed LF is presented. Section 5 explains the application of the proposed LF and the implementation of a specified platform. Section 6 discusses the applicability and effectiveness of the proposed LF in an EU project. This section also presents the preliminary results gained in this work and finally looks into possible future work.

## 2. Review of Literature

### 2.1. Digital Innovation Hubs

DIHs have the potential to become the main actor in transferring (digital) technologies within the reach of almost all industry sectors. DIHs attempt to provide a series of

support services (e.g., offering new digital services, promoting the existing services) and supplemental services (e.g., customer support, maintenance plans, product use enhancements, training, and skill development) to a wide variety of companies in their region and beyond [16]. As proximity is considered crucial, DIHs act as a doorway and a first regional point of contact that can offer multiple concrete and tangible forms of assistance in TT. Predominantly, DIHs provide such support and supplemental services either independently (through relying on their involved stakeholders) or through collaboration with networked DIHs (see Figure 1). The stakeholders of DIHs strive to identify the needs of companies and respond to them with access to the related services and solutions provided. It goes to the heart of DIHs' mission to meet the related needs of companies and create added value for them [10,15,17]. Even though DIHs have provided various useful supports and services, the portfolios, types, and features of their services still require to be properly defined, distinctly organized, and appropriately introduced to be then readily identified and extensively used/experienced by their customers (companies). Taking this point into account, Table 1 summarizes the main stakeholders, services, and benefits of DIHs.

Almost all businesses can benefit from the main supports and services listed in Table 1, whether a small local company or a large global enterprise. Particularly, the services for learning and skill development not only have a direct impact on companies' productivity, performance, and success and also expand the knowledge base of all their employees, but also play a significant role in increasing the competitiveness, credibility, and business growth of DIHs as well. In an agreement with the interested companies, DIHs can, for example, provide various training programs in many forms including but not limited to short-term training courses, coaching and mentoring programs, internships, tutorial sessions, workshops, seminars, webinars, showrooms, factory tours, commercials, interactive demonstrations, and events. These services can range from basic skills development (e.g., how to use software suites) to advanced and high-level courses in a university [18,19].

The provided supports and services for education, training, and learning (for companies) should cover almost the whole employment spectrum, need to be tailored to the specific needs of companies, and be defined based on an analysis of regional demands. For example, nowadays, digitally skilled laborers are in high demand. To meet this request, DIHs can pool the needs of their customers/companies and accordingly create and develop the specified training programs (e.g., vocational training, training the trainers) to match supply and demand efficiently. Additionally, to develop the knowledge and skills of laborers and promote their learning culture, DIHs can take advantage of different training methods, instructor-led training, technology-based learning, on-the-job training, use cases, films, and videos, to name but a few [16–19]. Table 2 summarizes the main educational, training, and learning supports and services that can be provided by DIHs for up-skilling and/or re-skilling the laborers of companies (as well as the mentors of training institutes and employees of public sectors).

It should be noted that the significant role of DIHs in digitalization, digitization, and (digital) TT was further highlighted during the global crisis caused by the COVID-19 pandemic. By way of illustration, the use of digital technologies and the fast shift to digital alternatives (e.g., online jobs, teleworking) enabled companies (particularly SMEs and small businesses) to mitigate the disruptions caused. This crisis enforced the need to accelerate the (digital) TT and digital transformation and to then make companies and businesses more agile, resilient, consistent, and flexible, underscoring that the role of DIHs is nowadays more important than ever [18].

**Table 1.** Main stakeholders, services, and benefits of DIHs.

| Main Features and Characteristics of DIHs | | |
| --- | --- | --- |
| Main Stakeholders | Main Supports and Services | Main Benefits for Companies |
| • Public sectors<br>• Government agencies<br>• Private sector<br>• Academia<br>• NGOs<br>• Chambers of commerce<br>• Industrial sectors<br>• Industry associations<br>• Large companies<br>• SMEs<br>• Start-ups<br>• Midcaps<br>• Corporations<br>• Extension agencies<br>• Accelerators<br>• Entrepreneurs<br>• Real estate agents<br>• Regional development agen-cies<br>• Incubators/accelerators<br>• RTOs<br>• Research centers<br>• Living labs<br>• Training institutes<br>• Knowledge communities<br>• Specialized experts<br>• Mentors<br>• Researchers<br>• Students<br>• Media<br>• Suppliers<br>• Investors | *Innovation Activities and TT*<br><br>• Awareness creation (e.g., about digital technologies, funding opportunities)<br>• Digitalization<br>  → Digital maturity assessment<br>  → Digital transformation road-mapping<br>• Developing technologies<br>• Providing lab facilities<br>• Access to infrastructure<br>• Providing innovative solutions<br>• Experimentation<br>• Testing and validation<br>• Concept validation and prototyping<br><br>*Learning and Skill Development*<br><br>• Training/mentoring<br>• Sharing knowledge, experiences, and good practices<br>• Developing skills<br>• Access to specialist expertise<br>• Advising (e.g., financing advice)<br>• Collaborative research on issues of common interest<br><br>*Business Development*<br><br>• Supporting and strengthening businesses<br>• Improving business/production process-es, products, or services<br>• Visioning and strategy development for businesses<br>• Commercialization<br>• Supporting incubation<br>• Networking<br>• Fostering relationships<br>• Brokering/matchmaking<br>• Connecting companies with investors<br>• Linking suppliers with customers | • Understanding the company's needs<br>• Identifying opportunities for digitization<br>• Adapting advanced technologies<br>• Transforming business<br>• Developing business<br>• Possessing significant know-how spanning<br>• Developing and validating innovative solu-tions<br>• Tackling innovation-related problems<br>• Assessing digital maturity<br>• Rapid access to consultative and expertise<br>• Access to related roadshows, workshops, and innovation camps<br>• Access to Funding and investor readiness services<br>• Access to learning channels<br>• Access to new knowledge and information<br>• Access to tailored help and advice<br>• Access to experimentation environments<br>• Access to living labs for validating new business/products<br>• Trying co-creation and synergy capture<br>• Prototyping, testing, and implementing the solutions<br>• Learning from experimenting<br>• Being mentored about the issues such as trend analysis, business model development, value-chain creation, market assessments, internationalization<br>• Training the workforce to be able to deal with the newly digitized processes, services, and products<br>• Developing skills<br>• Reducing risks |

### 2.2. Technology Transfer

Broadly speaking, TT refers to the process of conveying discoveries into products or services to be sold and/or developed by companies [20,21]. This means, TT focuses on the process of disseminating inventions, materials, scientific outcomes, data, designs, software, technical knowledge, skills, know-how, methods of manufacturing, and other related profit motives (for various reasons and at different development stages). As TT is about transforming ideas into opportunities, it can be considered an intrinsic part of the technological innovation process. An effective TT can guarantee the expansion of opportunities for innovation, allowing companies to concentrate more on market develop-

ment and profit generation without involvement in (all stages of) technology creation and development [22].

**Table 2.** Main education, training, and learning services that can be provided by DIHs.

| Main Services of DIHs Relate to Education, Training, and Learning | |
|---|---|
| • Raising awareness<br>• Professional education<br>• Training (e.g., quality, safety, sale)<br>• Mentoring and coaching<br>• Internship<br>• Traineeships and apprenticeships<br>• Exchanging curricula and training material<br>• Access to the latest knowledge<br>• Exchanging knowledge<br>• Developing competencies<br>• Testing and experimenting | • Hand and soft skills development<br>• Consulting<br>• Networking and collaboration<br>• Defining the research and innovation priorities<br>• Access to the latest trends, technologies, and innovations<br>• Access to expertise<br>• R&D<br>• Road-mapping<br>• Assessment<br>• Group discussion and activities<br>• Role-playing |

TT is, by nature, a complex undertaking that can even involve many elements such as scientific, non-scientific, technological, and non-technological factors and also many different stakeholders. Therefore, to perform the process of TT appropriately, the literature suggests different types of processes. For instance, [22] proposes eight steps to be followed, namely, the discovery of novel technologies at universities and/or research institutions, disclosure, technology evaluation, intellectual property protection, marketing, licensing, product development, and public use and financial returns. Another study [23] proposes a qualitative and linear model that is composed of six phases including invention disclosures, patent applications, technology licenses executed, technology licenses yielding income, technology royalties, and start-up companies that can use the transferred technology and consequently the creation of jobs and profit. This model may not take into account certain external factors, such as market demand, regulatory state policies, and environmental aspects. The proposed TT process by [24] involves four main phases. In this process, the first phase is innovation, where the idea is created, and the technology is developed. The second phase is static validation, which involves experimentation to explore the basic concepts of technology to solve the identified problems. The third phase is dynamic validation, which evaluates the technology through asking related questions and finding suitable answers. The fourth phase releases the technology for wider use when it is discerned as a useful and usable means.

The literature [25,26] shows that not only the process of TT but also its enhancement can be supported by DIHs, both locally and globally. Furthermore, the education, training, and learning services provided by DIHs plus their output knowledge (e.g., guidance, procedures, lessons learned, templates) can also be used (directly or indirectly) for creating, developing, and transferring (digital) technologies to the companies. The fact is that the process of TT can be performed in different ways, such as (a) through training the workforce (bringing wider access to trained people who can then further develop and exploit the technology and also develop new products, processes, applications, materials, or services), (b) licensing patented intellectual property to corporations, (c) publishing the results of investigations and experiments, (d) developing the relationships with industry and community participants, etc.

In sum, in the process of TT, the created or developed technology (and most likely its related technical knowledge, designs, materials, and inventions) at one organization (e.g., a DIH) will convey to other organizations (e.g., companies, training institutes, public sectors, and even other DIHs), typically for the purpose of commercialization, development, and progression [27,28]. DIHs are basically specialized for creating, developing, and

transferring particular technologies. It should be noted that, in this context, the process of TT mostly focuses on digital transformation, which incorporates digital and computer-based technologies and applications into a company's products, processes, services, and strategies. Although providing and delivering all types of demanding services and support is an enormously huge and complicated task, collaboration and networking with other DIHs not only can facilitate this task but also can enable a DIH to provide more customized and personalized solutions [13,15].

*2.3. Knowledge Creation*

Knowledge creation as a process refers to the activities and initiatives employed towards the generation and formation of new concepts, approaches, methods, techniques, products, services, and ideas that occur through interactions and can be used for the benefit of entities [29]. Knowledge creation is a spiraling process of interactions between two types of knowledge, explicit and tacit knowledge. To understand how the interactions lead to the creation of new knowledge, it is necessary to identify the spirals' various phases or edges' main characteristics. The integration of explicit and tacit knowledge makes it possible to conceptualize four conversion patterns [30]. These four knowledge conversion patterns include (a) socialization (from tacit to tacit knowledge), (b) externalization (from tacit to explicit knowledge), (c) combination (from explicit to explicit knowledge), and (d) internalization (from explicit to tacit knowledge) [31]. Figure 2 demonstrates the manifestation of four knowledge conversion patterns and their relation/position to the spiral of organizational knowledge creation, addressing the direction in which the knowledge evolves. The purple arrows are related to the individual to inter-organizational direction. The orange arrows are related in the opposite direction, from an inter-organizational to an individual focus.

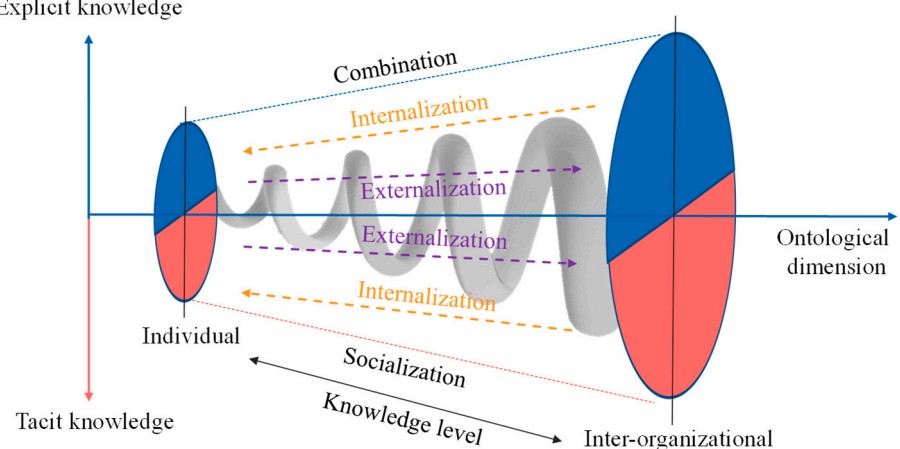

**Figure 2.** Four knowledge conversion patterns.

In the following, the four knowledge conversion patterns are explained in brief.

Socialization refers to sharing of tacit knowledge between individuals [30]. Sharing of tacit knowledge can be made through, for example, face-to-face communication and sharing experiences (e.g., observation, imitation, and practice). That is, socialization is characterized by direct interactions between individuals, where they share their emotions, feelings, and mental models. Thus, there should be factors such as commitment, trust, and love [32].

Externalization is the process of articulating tacit knowledge into explicit knowledge. When tacit knowledge is expressed/translated into explicit knowledge, the knowledge will be crystallized. In that account, it can be shared with others, and it becomes the basis of new knowledge. During the process of communication among individuals and groups, people exchange their beliefs and opinions as well as learn how to better articulate their thinking through instantaneous feedback and the simultaneous sharing of ideas [30].

Combination is the process of transforming explicit knowledge into more complex and systematic sets of explicit knowledge via, for example, physical and virtual communication or collective interactions. Therefore, such social interactions help to disseminate the new explicit knowledge among entities [30].

Internalization is the process of embodying explicit knowledge into tacit knowledge through practices and actions. Thus, in this process, individuals can gain knowledge through, for example, learning through observing, learning through doing, on-the-job training, and face-to-face meetings. Indeed, internalization can transfer the explicit knowledge of an organization or a group to individuals [32,33].

It should be noted that the tacit and explicit knowledge that is created through the spiral cycle presented in Figure 2 can be used for learning purposes by individuals and organizations (e.g., DIHs) [33,34].

### 2.4. Knowledge Development

Generally, knowledge development refers to the process of obtaining the most current and best-created knowledge and then reviewing, revising, and adding value to it, and also making the most of it [35,36]. Ref. [37] declares that knowledge development is an iterative and continuous process that fulfills at different levels (e.g., strategic, operational, and tactical) and increases the understanding of users (e.g., individuals and companies). To organize and utilize the developed (tacit and explicit) knowledge that comes from different sources and also to show their relationships in a meaningful manner, knowledge modeling can be used as a powerful and effective means [38]. Given the above, we propose a knowledge development model (see Figure 3) for supporting TT that stands on two main processes/cycles:

- Knowledge creation process/cycle: In this process, the created tacit and explicit knowledge (illustrated in Figure 2) initially helps to identify the related problems in the scope to be studied (step 1). Then, conducting the research and experiment can help increase our understanding of the scope of the study (step 2). Afterward, potential and promising solutions could be suggested based on the research and experiment carried out (step 3). Lastly, nominated solutions will be assessed to ensure that they can add value to the created knowledge adequately (step 4). This process can be accomplished (entirely or in part) in a DIH (network). After identifying certain solutions and creating the needed knowledge to a satisfactory or acceptable extent, the next process (training implementation) can be undertaken.

- Training implementation process/cycle: This process stands on the knowledge creation process, and the created knowledge can be used to design training courses as a starting point (step 1). The designed training courses can be then developed according to the objectives of the training program (step 2). Then, the training courses can be delivered by trainers and used by trainees (step 3). Lastly, through using some training assessment methods (e.g., formative assessment and summative assessment), the strengths and weaknesses of the training courses and trainees' performance will be identified and then adjusted/improved if needed (step 4). Such assessment methods could, for example, provide valuable indications about the quality, effectiveness, and efficiency of designed training courses as well as determine whether or not the training courses need change, modification, or development. Similarly, this process can be conducted (completely or partially) in a (network of) DIH(s).

The knowledge creation process/cycle and training implementation process/cycle can jointly crystallize the knowledge development model (in a DIH associated with different research projects or initiatives), as shown in Figure 3. This action can take place by following some steps, namely, setting a common goal (designing the knowledge development model), defining the role and contribution of each process/cycle, building effective and frequent communication channels and interactions between the processes/cycles, sharing needed resources, and monitoring the progress of the action. The stakeholders of DIHs can actively

play a role in organizing and managing this action. Moreover, the collaboration between networked DIHs can promote the related inputs, outputs, and outcomes.

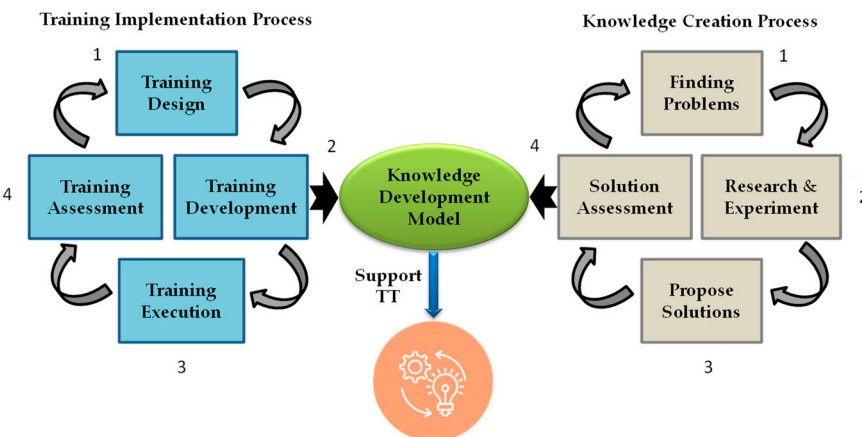

**Figure 3.** Proposed knowledge development model in a DIH.

The proposed knowledge development model can potentially facilitate and accelerate the process of TT in various ways and at different levels. By way of illustration, the knowledge development model can provide the required training materials (e.g., knowledge, information, data), experience, skills, know-how, and directions toward achieving successful TT to companies.

### 3. Methodology for Designing and Developing Project Ideas

This section describes the proposed waterfall hybrid methodology that was followed for the design and development of project ideas.

The proposed methodology and the learning framework were collaboratively produced by the partners of EU projects (that are introduced in the Discussion Section), resulting from several rounds of group discussion. As shown in Figure 4, the proposed methodology is called 'hybrid' because it encompasses two main levels that are explained in the following in brief.

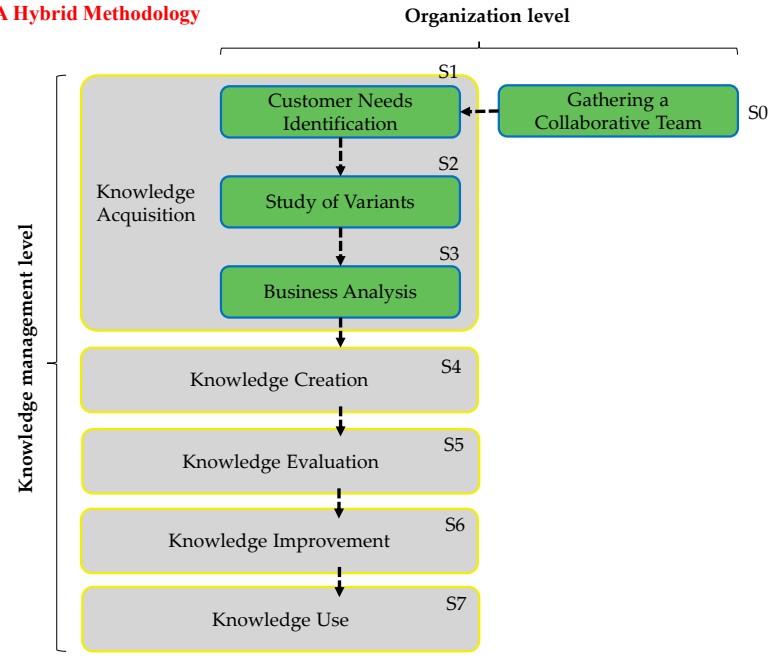

**Figure 4.** Proposed waterfall hybrid methodology for designing and developing project ideas.

A.   Organization level includes the four first stages of the methodology (S0–S3) and refers to the process of establishing the project ideas during the group discussions of the collaborative team (partners of the project and target stakeholders). This comprises steps of gathering a collaborative team (S0), identifying the customers' needs (S1), studying variants (S2), and analyzing the business model (S3).

B.   Knowledge management level has some overlapping with the three stages (S1–S3) of the organizational level. It also includes the other stages (S4–S7) of the methodology and refers to the process of managing the knowledge used for creating the project ideas. Thus, the knowledge management level contains steps of knowledge acquisition (S1–S3), knowledge creation (S4), knowledge evaluation (S5), knowledge improvement (S6), and knowledge use (S7).

The eight main stages of the waterfall hybrid methodology are briefly explained in the following:

A.   Organization level:

- Gathering a collaborative team (S0) refers to the plenary meetings that the partners of the project attended to exchange their ideas, information, and findings. This is an effective approach to communicating with partners, distributing information, discussing issues, and making decisions about different issues (e.g., variables and key performance indicators) to be considered in the project. This stage began before carrying out this work.
- Identifying the customers' needs (S1) refers to identifying (a) the potential and target customers (e.g., companies), (b) what they want and need, and (c) when they are set to interact with the service providers. Therefore, in the plenary meetings, the partners tried to identify the customers' needs through adopting various data collection methods and strategies, including conducting a deep literature review, customer surveys, customer interviews, social listening, analyzing the findings, and conducting several (partners) group discussions.
- Studying variants (S2) is a crucial stage in project development, as it helps to identify potential technical, organizational, economic, and financial challenges that may arise during the project implementation. These studies should consider both qualitative and quantitative requirements to ensure the successful resolution of problems that may arise during the project lifecycle. In this stage, it is important to consider the availability of needed resources, such as personnel and equipment. The study of variants should also take into account the influential factors that may affect the project (e.g., legal or regulatory requirements, social or environmental factors, or any other relevant issue).
- Business analysis (S3) refers to identifying the assessment criteria to be used in evaluating the business evaluation methods. The assessment criteria rely on technical variables identified in the prior stage and are evaluated from three perspectives, namely, financial and/or business, economic, and social. The financial and/or business perspective focuses on the funds invested in the project and assesses the financial feasibility of the project. This perspective also assesses the potential profitability and returns on investment of the project. The economic perspective considers the priorities of the economy and assesses the economic impact of the project. In addition, this perspective assesses the potential contributions of the project to job creation, economic growth, etc. The social perspective seeks to articulate the analysis of welfare which could define the social groups that benefit most from this methodology. This perspective assesses the potential social impact of the project and evaluates its potential to improve social welfare, such as improving access to education and training services and promoting social inclusion.

B.   Knowledge management level:

- Knowledge acquisition (S1–S3) refers to the process of extracting and selecting information (to better model it according to the collected interpretation). It also includes a formal structuring of the knowledge.
- Knowledge creation (S4) refers to borrowing some of the existing content from the literature and integrating it with the actors of this methodology/process (project ideas creators), information, and experiences to construct new knowledge through a dynamic, interactive, and collaborative process.
- Knowledge evaluation (S5) refers to assessing the applicability and effectiveness of the created knowledge and finding out its strengths and weaknesses.
- Knowledge improvement (S6) refers to the process of overcoming the detected weaknesses (in S5) and taking the needed action to improve the quality of knowledge.
- Knowledge using (S7) refers to transferring and applying the created knowledge (e.g., prototype implementation).

It is noteworthy that the proposed waterfall hybrid methodology in this work simply conceptualizes and mentally visualizes the main stages that need to be considered in the creation, establishment, and development of a project idea implementation. The methodology provides some hints and directions to developers and researchers who deal with this challenge. Additionally, the methodology is not static. It means that the number of levels, number of stages, and order of stages can be increased or decreased, according to the objectives, requirements, and conditions of the initiative (project idea creation).

## 4. Proposed Learning Framework

Taking the above methodology characteristics into consideration, a general LF for DIHs (LF-DIHs) was proposed. The LF-DIHs represents a mixed approach and follows evidence-based practice (which is defined as a problem-solving and decision-making approach). The LF-DIHs intends to help DIHs and instructors to (a) align learning goals within pedagogical activities, (b) create a motivating and inclusive environment for training and learning purposes, and (c) incorporate appropriate assessment mechanisms into training and learning activities. Furthermore, the LF-DIHs can provide guidance for training preparation, training execution, and training development. As mentioned earlier, the LF-DIHs is a general framework and contains a number of specific and general components that work together to support the main goal of a DIH to ensure an effective TT through digitalization. The LF-DIHs is also dynamic by nature as the instantiated methodology. Thus, it should be appropriately adapted and customized according to the conditions, requirements, and goals of the target DIH.

As depicted in Figure 5, the LF-DIHs instantiates and stands on the waterfall hybrid methodology. It comprises two levels (organizational and knowledge management), which are presented earlier, and also two layers. The layers are created on top of the knowledge management level. The first layer is called the technology layer (blue part). This layer consists of four main steps, namely, technical implementation, technology assessment, technology adaptation, and TT. This layer represents the lifecycle of technology creation in DIHs and technology transmission to companies. The second layer is named the training layer (orange part). Similarly, this layer involves four main steps, namely, training design, training assessment, training development and improvement, and training execution. This layer displays the lifecycle of training creation in DIHs and training delivery to companies (or training centers). Both layers end with the final step (exploration). The exploration step might be a termination of the process or lead to ex-post evaluation [39], which may result in an entire reformulation of the project idea or product.

It should be added that the training layer can lead to lifelong learning, as shown in Figure 5. In this work, lifelong learning refers to the ongoing, voluntary, and self-motivated pursuit of knowledge and skills throughout a person's entire life. It is the concept of continuous learning beyond formal education, extending beyond the traditional boundaries of classrooms and institutions (that can take place in, for example, DIHs). Lifelong learning

recognizes that learning is a lifelong process that occurs in various contexts and at different stages of life.

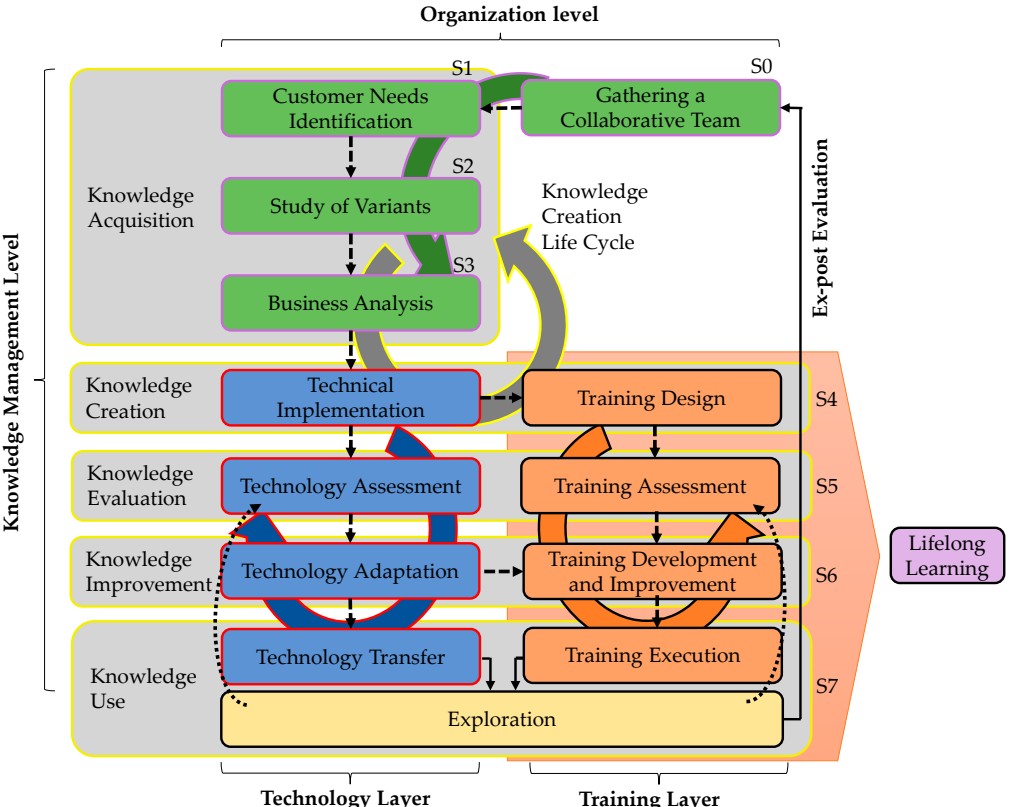

**Figure 5.** Proposed LF-DIHs.

## 5. Application of Learning Framework Digital Innovation Hubs and Implementation of Lifelong eLearning Platform

This section presents the way in which the LF-DIHs was evaluated, improved, and used, addressing its application to a Lifelong eLearning Platform (LeLP). In this study, the LeLP refers to an online platform that facilitates continuous learning and skill development throughout a person's lifetime. It is designed to provide individuals with access to a wide range of educational resources, courses, and learning opportunities anytime and anywhere. Indeed, LeLP leverages technology to deliver learning materials, interactive activities, assessments, and collaboration tools to support self-paced and personalized learning experiences.

The application of the LF-DIHs to LeLP has been considered, tested, and supported by two EU projects, (a) the ENHANCE project (http://eplus-enhance.eu/, accessed on 28 March 2023) and (b) the PRODUTECH project (http://www.produtech.org/produtech-dih-platform, accessed on 28 March 2023). In this work, we merely report the related results that are gained and released as part of the ENHANCE project. The PRODUTECH project that plans to implement an E-DIH will then explore the results of this work. This part of the work deals with S0 of the LF-DIHs.

ENHANCE is co-funded by the Erasmus + Programme of the European Union. The project contributes to strengthening the skills and training expertise of Tunisian and Moroccan universities in three targeted topics, namely, maintenance engineering, production engineering, and quality engineering (which refer to MPQ) for inciting and assisting both partner countries' transition to the Industry 4.0 era [40]. One of the main objectives of the project focuses on developing a learning framework and a LeLP.

In order to test the applicability and effectiveness of the LF-DIHs to the ENHANCE LeLP, we first needed to analyze the adequacy of the LF-DIHs to the ENHANCE use

case. Thus, to verify the adequacy of the LF-DIHs, in some online and face-to-face plenary group meetings that the partners had together, they had several discussions and arguments about the adequacy, applicability, and effectiveness of the LF-DIHs for the considered use case. In these meetings, they shared their views and opinions around the following related questions:

- Is the quality of the LF-DIHs good enough to be accepted?
- Is the LF-DIHs relevant to the considered use case?
- Is the LF-DIHs appropriate for the considered use case?
- Is the LF-DIHs fit the considered use case?
- Is the LF-DIHs useful for the considered use case?
- Will the LF-DIHs be successful in producing the desired result in the considered use case?

Through critically assessing the features and capabilities of LF-DIHs and the needs of the considered use case, the partners eventually came to the conclusion that the existing (theoretical) evidence makes a convincing impression of the adequacy, applicability, and effectiveness of the LF-DIHs for the considered use case. That is, they decided to keep the LF-DIHs as it is until reaching the results of testing the LF-DIHs on the use case. The idea was to use the concepts, components, and principles of the LF-DIHs to analyze and interpret the data, assessing how well the LF-DIHs addresses the research question or objective for the test of the use case. Analyzing the obtained results not only can show whether the LF-DIHs is adequate, applicable, and effective for the use case but also helps to identify the strengths and weaknesses of the LF-DIHs. Given that, the partners can make the needed decisions and actions accordingly. This part of the work deals with the S1 of the LF-DIHs. S2 is covered in Table 1, and S3 is covered by the explanation presented in two paragraphs between Tables 1 and 2. S3 involves the analysis of business processes, systems, and strategies to improve efficiency, effectiveness, and overall performance.

The application and adaptation of the LF-DIHs to the LeLP have direct and indirect relationships with the implementation of the LeLP and its specified objectives, components, features, and functions of the platform. Given that, the following sub-sections briefly explain the way in which the objectives, components, features, and functions of the LeLP are defined, addressing which steps had been taken for the implementation of the LeLP. It should be added that the following actions and Sections 5.1–5.3 took place before practically launching the LeLP and using the LF-DIHs.

### 5.1. Defining the Objectives of the Lifelong eLearning Platform

The LeLP is specifically designed to be used as a central place for providing a broad range of services. For example, it provides users with access to search engines, personalized home pages, available information, chat forums, and email services [41]. In that account, the following three objectives for the LF-DIHs are collaboratively defined by the partners and stakeholders of the project to help set the goals in a way that all related activities lead to one single direction:

- Defining Training Activities refers to specifying the missions, plans, programs, exercises, practices, and other activities that could improve learners' qualifications, skills, and knowledge.
- Assessing Competences refers to assessing learners' strengths and weaknesses in connection with the requirements for their studies and current/future jobs.
- Designing Training Curriculum focuses on creating, improving, and organizing the needed training courses that should be provided by target companies/universities. It also deals with what will be taught, who will be taught, and how it will be taught.

### 5.2. Identifying and Selecting the Potential Components and Features to Be Used on the Lifelong eLearning Platform

To identify the potential components and features which can be used on the LeLP, the organizational structures and specifications of 15 cases of the collaborative learning

community (e.g., Wikipedia, Digg, Yahoo! Answers, SETI@home, Scratch, Galaxy Zoo, Foldit, Applications of the Delphi method, Climate Colab, Assignment Zero, Donation-Coder, Experts Exchange, Waze, Makerspaces, and SAP Community Network) were first reviewed and analyzed, and then the potential components and features were nominated by the partners [41]. In order to evaluate the adequacy of nominated components and features for use on the LeLP, they were accordingly adapted and addressed in 90 questions (in a questionnaire) under 11 dimensions of collaboration, namely, organizational, environmental, admission, social, functional, economical, technological, structural, behavioral, learning assessment, and performance assessment dimensions. Thirteen evaluators (partners and stakeholders who are professors and experts in the field of industrial engineering, computer engineering, and computer science) participated in this step of evaluation as a focus group. The evaluators were asked to leave their feedback about the nominated components and features by using the considered five-point Likert scale. The results of this evaluation—which are presented in [41]—helped us to select those components and features that received the highest percentage of popularity/acceptance by evaluators. The selected components and features were then used on the LeLP.

*5.3. Determining the Main Functions of Lifelong eLearning Platform According to the Steps of Learning Framework for Digital Innovation Hubs*

Considering the three defined objectives and the requirements of the platform, the following six functions are collaboratively defined for the LeLP:

- Dynamic training design (followed in S4 of the LF-DIHs),
- Training program generation (followed in S4 of the LF-DIHs),
- Training quality assessment (followed in S5 of the LF-DIHs),
- Training content improvement (followed in S6 of the LF-DIHs),
- Training execution support (followed in S7 of the LF-DIHs), and
- User and information management (followed in S7 of LF-DIHs).

To evaluate the adequacy of the determined functions, a questionnaire containing 34 questions was designed and delivered to the partners (who participated in the previous step of evaluation), addressing the most important operational and execution aspects to be considered for the platform. The results of this evaluation provided very good indications of the importance, strength, capabilities, requirements, and constraints of the platform functions [41].

*5.4. Evaluating the Implementation of the Main Functions of the Lifelong eLearning Platform*

By taking into account the results of (a) defining the objectives of the LeLP, (b) identifying the components and features of the LeLP, and (c) determining the main functions of the LeLP, the implementation of the main functions of the LeLP was evaluated theoretically and conceptually by the partners and stakeholders in plenary meetings. This evaluation focused on judging whether or not the components and tools of the LeLP can adequately support the main functions of the LeLP. That is, those components and tools of the LeLP that show signs and symptoms of applicability for supporting one or some functions (of the LeLP) are marked with (X) in Table 3. It should be pointed out that at this stage of LeLP development, six main components and tools are suggested for LeLP by partners, which are addressed in Table 3 and presented in the next section.

Table 3 gives an overview of potential relationships and interactions among the main components and tools of the LeLP and its functions based on the common agreement of evaluators. For example, as shown in Table 3, two components and tools of the LeLP, namely learning activities, syllabuses and authoring tools, can be applied for function 1 (dynamic training design).

**Table 3.** Results of evaluating the implementation of the main functions of the Lifelong eLearning Platform.

| Functions of LeLP | Components and Tools of LeLP | | | | | |
|---|---|---|---|---|---|---|
| | LearningActivities Syllabuses | Knowledge from DIHs | Portal Services | Authoring Tool | Learning Management System | Learning Record Storage |
| 1. Dynamic training design | x | | | x | | |
| 2. Training program generation | | | x | | x | |
| 3. Training quality assessment | | | x | | x | x |
| 4. Training content improvement | | x | | x | | |
| 5. Training execution support | | | x | | x | x |
| 6. User and information management | | | x | | | |

*5.5. Implementation of Lifelong eLearning Platform through Instantiation of Training and Technology Layers*

This section explains how the LeLP and its components and tools can be implemented through the instantiation of training and technology layers. In order to use the training and technology layers for the training purpose, the following three steps should be first taken into account:

(a) Step 1: Identifying Potential Instructors refers to identifying qualified instructors or training potential trainers who are interested and able to deliver the training syllabuses and courses.

(b) Step 2: Clarifying Training Purposes and Role Expectations refers to providing related guidance and detailed information about, for example, what are the objectives of the training syllabuses and courses, how to meet them, what are the tasks and activities, and how to perform them.

(c) Step 3: Bringing About the Required Infrastructure and Components refers to providing the basic physical systems and a set of tools and components that support the process of implementation, use, and delivery of training syllabuses and courses.

In the following, a brief description is provided for each proposed component and tool of the LeLP.

- Learning Activities Syllabuses are a set of learning documents that provide useful and practical information about specific academic courses and/or classes. Generally, the syllabuses provide an overview or summary of the curriculum to be delivered, and they can serve as a guide to a course and what will be expected of the learner during the course. These syllabuses may include the expectations, responsibilities, course policies, rules, regulations, required texts, and schedule of assignments.

- The Training and Learning Portal of ENHANCE is a specified web-based platform (and historically used to refer to a gateway for a World Wide Web) that collects information from different sources (e.g., online forums, search engines, and emails) into a single user interface and presents users with the most relevant information for their training and learning.

- Knowledge from DIHs refers to the facts, truth, awareness, data, information, and findings that are identified, acquired, created, shared, and/or developed by DIHs for different purposes (e.g., education, training, and learning).
- Moodle is an open-source 'learning management system' that (in addition to content management) allows to build and upload e-learning content, deliver it to learners, assess the content continually, track learners' progress, and recognize their achievements. Moodle also provides a central space on the portal where learners can access a set of tools, resources, and courses anytime and anywhere. Moodle helps to conceptualize the various courses, course structures, and curricula, thus facilitating interaction and communication with online learners (for example, in discussion forums).
- xAPI is an e-learning software specification that allows learning content and learning systems to speak to each other in a manner that records and tracks all types of learning experiences. xAPI introduces the standards that define and adjust the tracking, sharing, and storing of learners' learning performance across the portal. With xAPI, authorities can track (almost) anything that the learners do. Learning experiences are recorded in learning record storage.
- Learning Record Storage (LRS) is a data storage system that serves as a repository for learning records collected from connected systems where learning activities are conducted. The Learning record storage is the heart of the xAPI ecosystem and assists in receiving, bringing together, storing, and returning learning records and xAPI statements where the learning activities are conducted (e.g., in the portal). Every other tool which sends or retrieves learning activity data will interact with the learning record storage as the central store.
- Authoring Tools are software programs that assist instructional designers in creating online courses and related content/knowledge and publishing them in desired formats. Authoring tools also enable designers to customize lessons, tutorials, and digital content, using various forms of media (e.g., text) and interactions. Authoring tools can organize and deploy content or upload it to a learning management system (e.g., Moodle). Authoring tools can also help in creating software simulations, gamification, and building questions.

Figure 6 simply visualizes the implementation and use of the LeLP through the instantiation of training and technology layers.

The idea is that the training layer supports and facilitates the implementation and use of four components and tools of the LeLP, namely, authoring tools, LRS, xAPI, and Moodle. The other components and tools (learning activities syllabuses, training and learning portal of ENHANCE, and knowledge from DIHs) should be supported and facilitated directly via the technology layer. In the second one, the preparation of learning activity syllabuses (e.g., creation, development, evaluation, and updating) will be supported through the interaction and incorporation of the steps of technology layers and also simplified using a technological pedagogic content knowledge approach [42]. In this respect, the partners provided three training programs, focusing on production, maintenance, and quality engineering processes, which represent the key industrial business processes that particularly need attention, investment, and improvement in Tunisia and Morocco.

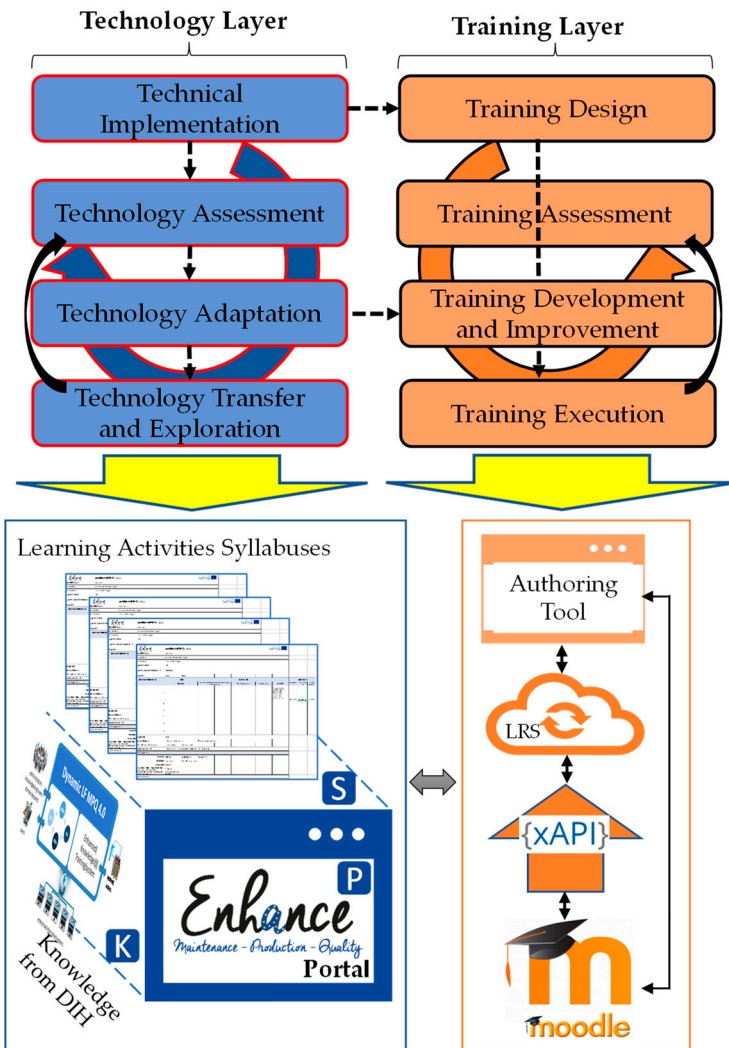

**Figure 6.** Implementation and use of LeLP through the instantiation of training and technology layers.

## 6. Discussion

In this section, we discuss the applicability and effectiveness of the LF-DIHs to the LeLP from the partners' and stakeholders' points of view.

### 6.1. Evaluating the Applicability of Learning Framework for Digital Innovation Hubs to Lifelong eLearning Platform

The applicability of the LF-DIHs refers to its suitability and relevance for addressing the objectives of the LeLP (presented in Section 5.1). It is a measure of how well the LF-DIHs can be applied and utilized in the ENHANCE use case to achieve desired outcomes and results. The partners and stakeholders decided to evaluate the applicability of the LF-DIHs before and after implementing the LeLP because they believe that the more data they collect through multiple evaluations, the more valid and reliable evidence they can provide. Therefore, the partners and stakeholders first proceeded with the evaluation of the applicability of the LF-DIHs before implementing the LeLP through taking the following steps:

- Clarifying objectives: In this step, the partners and stakeholders tried to understand and clarify the specific objectives they are trying to address and achieve. They clearly defined the scope and context in which the LF-DIHs will be applied to the LeLP. This helped them assess whether the LF-DIHs aligns with their needs.

- Assessing the alignment of the LF-DIHs with objectives: In this step, the partners and stakeholders theoretically evaluated how well the LF-DIHs addresses the objectives at hand. They examined the concepts, principles, steps, components, and tools outlined in the LF-DIHs and determined if they are relevant and applicable to the LeLP.
- Considering the context and environment: In this step, the partners and stakeholders assessed the compatibility of the LF-DIHs with the ENHANCE use case's context, industry, and culture. They considered factors such as the size of the ENHANCE use case, the nature of the needed operations, and the maturity of the related processes. Then, they tried to ensure that the LF-DIHs can be effectively implemented within the ENHANCE use case context.
- Evaluating feasibility and resource requirements: In this step, the partners and stakeholders determined the feasibility of implementing the LF-DIHs in terms of resources, skills, and infrastructure. They assessed whether the ENHANCE use case has the necessary capabilities, expertise, and resources to adopt and sustain the LF-DIHs. They also considered the costs and potential benefits associated with implementation.
- Seeking expert opinions and feedback: In this step, the partners and stakeholders consulted with external experts, practitioners, and professionals in the field who have experience with the application of such a framework. The partners and stakeholders then gathered their opinions, feedback, and insights on the applicability of the LF-DIHs to the specific situation of the LeLP. It should be added that their expertise helped partners and stakeholders assess the suitability of the LF-DIHs.

The preliminary results gained through evaluating the applicability of the LF-DIHs to the LeLP are satisfactory. The results show that the LF-DIHs has a high potential to be applied to LeLP, given that—at the stage of closing this study—the partners are applying the LF-DIHs to the LeLP. However, when the LeLP is completely implemented and a considered pilot test evaluates the effectiveness, challenges, and outcomes of the LF-DIH, the partners and stakeholders will then process a further applicability evaluation (evaluating after implementing the LeLP) through taking the following steps:

- Data availability and quality: In this step, the partners and stakeholders will evaluate the availability and quality of the data required to feed into the LF-DIHs. For doing so, they will consider factors such as data accuracy, representativeness, relevance, and timeliness. They will then evaluate whether the available data aligns with the requirements of the LF-DIHs and whether any data gaps may affect the applicability of LF-DIHs.
- Compare the LF-DIHs' outputs with real-world data: In this step, the partners and stakeholders will compare the outputs or predictions of the LF-DIHs with real-world data or observations, if available. They will assess the agreement or discrepancy between the LF-DIHs' outputs and the observed outcomes. They will also consider the level of accuracy, precision, and reliability demonstrated by the LF-DIHs in reproducing real-world use cases.
- Sensitivity and robustness analysis: In this step, the partners and stakeholders will perform a sensitivity analysis to assess the LF-DIHs' sensitivity to changes in input parameters or assumptions. They will vary the input parameters within a reasonable range and observe the impact on the LF-DIHs' outputs. This analysis helps partners to understand the robustness and stability of the LF-DIHs' results.
- Expert evaluation: In this step, the partners and stakeholders again will seek expert opinions and insights from domain experts who have experience and knowledge in this specific field of application. Undoubtedly, experts can provide valuable perspectives on the LF-DIHs' applicability, potential biases, and limitations based on their practical experience and understanding of the system or process being modeled.
- Users' feedback: In this step, the partners and stakeholders will also gather feedback from relevant users who will be affected by the LF-DIHs' application and use the LeLP. The partners and stakeholders will consider their perspectives, concerns, and expecta-

tions regarding the LF-DIHs' applicability. Their feedback will be then incorporated into the evaluation process.

*6.2. Evaluating the Effectiveness of the Learning Framework for Digital Innovation Hubs to Lifelong eLearning Platform*

The fact is that evaluating theoretically the effectiveness of the LF-DIHs was an arduous and daunting task for the evaluators (partners and stakeholders) because there was a lack of practical and evidence-based results. However, the evaluation that they made enabled them to make predictions about the effectiveness of the LF-DIHs based on what they could observe. In this step of evaluation, the partners and stakeholders assessed the effectiveness of the LF-DIHs, aiming to judge the degree of its success in achieving the goals of the LeLP (mentioned in Section 5.1). This task is concerned with comparing (at this stage, theoretically) the inputs (concepts, principles, steps, features, and capabilities) of the LF-DIHs with the desired outputs that they can make. In some plenary group meetings that the partners and stakeholders had together, they had several discussions and arguments about the effectiveness of the LF-DIHs to the LeLP. In these meetings, they shared their views and opinions about the following related questions:

- Can the LF-DIHs produce a deep and vivid impression of its effectiveness?
- Can the LF-DIHs bring about an effect on the LeLP and its components and tools?
- Can the LF-DIHs produce the desired results and success of the LeLP?
- Can the LF-DIHs be applied to the LeLP with minimum financial, physical, and human resources?

Again, through critically assessing different aspects of the LF-DIHs and its features and capabilities from an effective point of view, the partners and stakeholders eventually drew the conclusion that the existing (theoretical) evidence makes a convincing impression of the effectiveness of the LF-DIH for applying to the LeLP. Therefore, they agreed to keep the LF-DIHs as it is and use it for the LeLP temporally. After practically implementing the LeLP, they will then proceed with the second round of effectiveness evaluation through taking the following steps:

- Review framework documentation: In this step, the partners and stakeholders will review the documentation and guidelines associated with the LF-DIHs. They will again evaluate the expected purposes and outcomes that should be fulfilled by the LF-DIHs.
- Data collection: In this step, the partners and stakeholders will collect relevant data to assess the success of the LF-DIHs. This includes quantitative data (e.g., performance metrics) and qualitative data (e.g., user feedback). The partners will try to ensure that the data collected aligns with the criteria defined for evaluation.
- Performance evaluation: In this step, the partners and stakeholders will apply the LF-DIHs to the platform and use historical data to evaluate its performance. They will measure the outcomes achieved using the LF-DIHs and compare them against the defined criteria and objectives.
- User feedback and surveys: In this step, the partners and stakeholders will gather feedback from users who have experience with the LF-DIHs. The partners will conduct surveys or interviews to assess their satisfaction, usability, and perception of the LF-DIHs' effectiveness. The partners and stakeholders will then consider their suggestions for improvement or areas where the LF-DIHs may fall short.
- Comparative analysis: In this step, the partners and stakeholders will conduct a comparative analysis through benchmarking the LF-DIHs against alternative approaches or competing frameworks. They will then evaluate how the LF-DIHs compares in terms of efficiency, effectiveness, cost-effectiveness, or other relevant criteria. This analysis can provide insights into the relative strengths and weaknesses of the LF-DIHs.
- Expert evaluation: In this step, the partners and stakeholders will seek expert opinions and feedback from professionals or domain experts. The partners and stakeholders will engage them in evaluating the effectiveness of the LF-DIHs based on their expertise and understanding of the field.

- Iterate and improve: In this step, based on the evaluation findings, the partners and stakeholders will identify areas where the LF-DIHs can be improved or refined. They will incorporate feedback and suggestions from users and experts to enhance the LF-DIHs' effectiveness. Lastly, they will iteratively refine the LF-DIHs based on evaluation results and feedback.

If the results of the second round of effectiveness evaluation are acceptable, the LF-DIHs can then be considered for further use in the ENHANCE use case and tested for application in other use cases (e.g., the PRODUTECH project).

The fact is that the processes of evaluating and improving the LF-DIHs were accomplished simultaneously, meaning that wherever the evaluation showed that the LF-DIHs needs change, betterment, advance, and improvement, the partners and stakeholders attempted to make the required modifications toward bringing the LF-DIHs into a more valuable or desirable condition. The results of several modifications and improvements led to the current version of the LF-DIHs.

Despite this work facing some limitations (e.g., lack of practical application and incomplete assessment of implementation challenges), through the development and partial validation of the LF-DIHs and related solutions, the proposed hypothesis is validated to a reasonable extent. Therefore, it can be concluded that the process of (digital) TT to companies can be supported by DIHs when the existing or developed/adjusted version of the LF-DIHs is used as guidance. Accordingly, universities and DIHs in Tunisia and Morocco can use the LF-DIHs to design and deliver training courses that can facilitate and support the process of (digital) TT to local (interested) companies.

### 6.3. Future Work

It should be pointed out that the process of evaluation, improvement, and use of the LF-DIHs is still in progress. In future work, we will first launch the LeLP (which is now in design), then the well-prepared and developed LF-DIHs will be installed and implemented on it. Afterward, the users (e.g., DIHs, teachers, and students) will be asked to use and evaluate the usefulness and effectiveness of the LeLP, aiming to complete the process of knowledge evaluation (S5). The evaluation could be performed, for example, through providing online questionnaires that should be filled out by users. In the next stage, the LeLP and LF-DIHs will be developed (collaboratively) according to the feedback received from users. Ideally, the LF-DIHs will be then applied to similar cases and scenarios, aiming to identify what should be improved and make the needed decisions and actions to increase the applicability and effectiveness of the LF-DIHs. Finally, the results gained from all these proceedings will be disseminated.

**Author Contributions:** Conceptualization, J.S., M.Z. and A.A.; methodology, J.S., M.Z. and E.M.-J.; validation, J.S. and M.Z.; formal analysis, J.S. and M.Z.; investigation, J.S. and M.Z.; resources, R.J.-G. and N.M.; data curation, J.S., M.Z. and A.A.; writing—original draft preparation, J.S., M.Z. and E.M.-J.; writing—review and editing, J.S. and M.Z.; visualization, J.S., M.Z. and A.A.; supervision, J.S., R.J.-G. and N.M.; project administration, J.S., R.J.-G. and N.M.; funding acquisition, R.J.-G. and N.M. All authors have read and agreed to the published version of the manuscript.

**Funding:** This study was funded by Fundação para a Ciência e Tecnologia (project UIDB/00066/2020) and European Commission ERASMUS + through grant n°. 619130-EPP-1-2020-1-FR-EPPKA2-CBHE-JP ENHANCE (http://eplus-enhance.eu, accessed on 28 March 2023). It was also supported by the project PRODUTECH DIH through call DIGITAL-2021-EDIH-01 with submission number 101083487.

**Data Availability Statement:** Not applicable.

**Acknowledgments:** This study was supported by the Portuguese FCT program, Center of Technology and Systems (CTS) UIDB/00066/2020/UIDP/00066/2020.

**Conflicts of Interest:** The authors declare no conflict of interest.

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
