# Peer review of "A Learning Framework for Supporting Digital Innovation Hubs"

_computers, doi:10.3390/computers12060122_

Round 1

Reviewer 1 Report

Introduction introduces DIH but not the goal ot the article 

Section 3 does not explain the objective of the article and research questions related to the article and also resrearch methods used in the article are not explained - currtent section 3 explains "methodology" build in project(s) to which article is linked. Research methodology/methods of the article are not defined and explained.

In section 4 propsed learning framework is described (I assume this was the outcome of the research) but it is not explanied how the framework was verified.

Discussion looks as report from the project and  should not discuss the project(s) but the research goals (and finidings) of the article

Reviewer 2 Report

The paper proposes a learning framework to assist digital innovation hubs in providing proper education, training, and learning services that support the process of (digital) technology transfer to companies. The framework was designed, evaluated, and improved with the support of two EU projects, and it has immense potential for application to similar cases.

The article performs an adequate theoretical framework through a relevant and current literature review when discussing discusses the increasing demand for digital transformation and technology transfer and how digital innovation hubs can provide good opportunities to access the needed innovation, technologies, and resources at a higher level than other organizations that can generally access them. However, the paper notes that DIHs are still evolving, under research, and development. Transferring, adapting, and integrating (digital) technologies might impose various challenges and entail different organizational and technical changes to businesses and companies. Despite these challenges, digital transformation and technologies have enabled businesses and companies of all sizes to boost their competitiveness. The authors have borrowed some of the existing content from the literature and integrated it with their information and experiences to construct new knowledge through a dynamic, interactive, and collaborative process.

Figure 2 should be replaced with a better-quality one. Do the authors create this figure? If not, the source must be cited.

The paper describes a hybrid waterfall methodology that was followed to design and develop the proposed learning framework. The methodology and the learning framework were collaboratively produced by the partners of EU projects, resulting from several rounds of group discussion. A software development methodology is presented. However, the discussion group method is not clearly explained. How many participants? How many years of experience do they have? What is their graduation? See the Focus Group method. These questions are partially answered in the discussion section.

It is not provided a detailed discussion of the limitations of the proposed learning framework. Authors should consider their introduction in the manuscript.

The "Future Work" section is not numbered. If this is the conclusions section, it is suggested to organize the conclusion section much better.

Round 2

Reviewer 1 Report

The article is now much better